# On the Prospects of Incorporating Large Language Models (LLMs) in Automated Planning and Scheduling (APS)

**Primary Keywords:** *(1) Large Language Models; (2) Learning; (3) Planning*

## Abstract

Automated Planning and Scheduling is among the growing areas in Artificial Intelligence (AI) where mention of LLMs has gained popularity. Based on a comprehensive review of 126 papers, this paper investigates eight categories based on the unique applications of LLMs in addressing various aspects of planning problems: language translation, plan generation, model construction, multi-agent planning, interactive planning, heuristics optimization, tool integration, and brain-inspired planning. For each category, we articulate the issues considered and existing gaps. A critical insight resulting from our review is that the true potential of LLMs unfolds when they are integrated with traditional symbolic planners, pointing towards a promising neuro-symbolic approach. This approach effectively combines the generative aspects of LLMs with the precision of classical planning methods. By synthesizing insights from existing literature, we underline the potential of this integration to address complex planning challenges. Our goal is to encourage the ICAPS community to recognize the complementary strengths of LLMs and symbolic planners, advocating for a direction in automated planning that leverages these synergistic capabilities to develop more advanced and intelligent planning systems.

## Introduction

As a sub-field of Artificial Intelligence (Russell and Norvig 2003), Automated Planning and Scheduling (Ghallab, Nau, and Traverso 2004) refers to developing algorithms and systems to generate plans or sequences of actions to achieve specific goals in a given environment or problem domain. APS is a valuable tool in domains where there is a need for intelligent decision-making, goal achievement, and efficient resource utilization. It enables the automation of complex tasks, making systems more capable and adaptable in dynamic environments. Over time, APS has evolved from the early development of robust theoretical foundations to practical applications in diverse sectors like manufacturing, space exploration, and personal scheduling. This evolution underscores the versatility and critical significance of APS.

In parallel with advancements in APS, the development and proliferation of LLMs have marked a substantial leap in AI, particularly within computational linguistics. Evolving from early efforts in natural language processing (NLP), LLMs have undergone significant transformation. Initially focused on basic tasks like word prediction and syntax analysis, newer models are characterized by their ability to generate coherent, contextually relevant text and perform diverse, complex linguistic tasks. Trained on extensive text corpora, LLMs have mastered human-like language patterns. Their recent success in various NLP tasks has prompted efforts to apply these models in APS. There is a notable shift towards using language constructs to specify aspects of planning, such as preconditions, effects, and goals, rather than relying solely on traditional planning domain languages like PDDL.

This paper presents an exhaustive literature review exploring the integration of LLMs in APS across eight categories: Language Translation, Plan Generation, Model Construction, Multi-agent Planning, Interactive Planning, Heuristics Optimization, Brain-Inspired Planning, and Tool Integration. Table 1 provides the description for the eight categories. Our comprehensive analysis of 126 papers not only categorizes LLMs' diverse contributions but also identifies significant gaps in each domain. Through our review, we put forward the following position:

> ### Position Statement
>
> Integrating LLMs into APS marks a pivotal advancement, bridging the gap between the advanced reasoning of traditional APS and the nuanced language understanding of LLMs. Traditional APS systems excel in structured, logical planning but often lack flexibility and contextual adaptability, a gap readily filled by LLMs. Conversely, while LLMs offer unparalleled natural language processing and a vast knowledge base, they fail to generate precise, actionable plans where APS systems thrive. This integration surpasses the limitations of each standalone method, offering a dynamic and context-aware planning approach, while also scaling up the traditional use of data and past experiences in the planning process.

In the forthcoming sections, we delve into the background of LLMs and classical planning problem, accompanied by the identification of literature. This sets the stage for an in-depth exploration of the application of LLMs in APS, where we critically examine the strengths and limitations of LLMs. Our position on the emerging neuro-symbolic AI paradigm

| Category | Description |
| --- | --- |
| Language Translation | Involves converting natural language into structured planning languages or formats like PDDL and vice-versa, enhancing the interface between human linguistic input and machine-understandable planning directives. |
| Plan Generation | Entails the creation of plans or strategies directly by LLMs, focusing on generating actionable sequences or decision-making processes. |
| Model Construction | Utilizes LLMs to construct or refine world and domain models essential for accurate and effective planning. |
| Multi-agent Planning | Focuses on scenarios involving multiple agents, where LLMs contribute to coordination and cooperative strategy development. |
| Interactive Planning | Centers on scenarios requiring iterative feedback or interactive planning with users, external verifiers, or environment, emphasizing the adaptability of LLMs to dynamic inputs. |
| Heuristics Optimization | Applies LLMs in optimizing planning processes through refining existing plans or providing heuristic assistance to symbolic planners. |
| Tool Integration | Encompasses studies where LLMs act as central orchestrators or coordinators in a tool ecosystem, interfacing with planners, theorem provers, and other systems. |
| Brain-Inspired Planning | Covers research focusing on LLM architectures inspired by neurological or cognitive processes, particularly to enhance planning capabilities. |

Table 1: Comprehensive description of the eight categories utilizing LLMs in APS

is central to our discussion, highlighting its unique advantages over purely neural network-based (i.e., statistical AI) or symbolic AI approaches. Finally, we will discuss prospective developments, address potential challenges, and identify promising opportunities in the field.

## Background

### Large Language Models

Large language models are neural network models with upwards of $\sim$ 3 billion parameters that are trained on extremely large corpora of natural language data (trillions of tokens/-words). These models are proficient in interpreting, generating, and contextualizing human language, leading to applications ranging from text generation to language-driven reasoning tasks. The evolution of LLMs in NLP began with rule-based models, progressed through statistical models, and achieved a significant breakthrough with the introduction of neural network-based models. The shift to sequence-based neural networks, with Recurrent Neural Networks (RNNs) and Long Short-Term Memory (LSTM) networks, marked a notable advancement due to their capability to process information and context over long sequences. Shortcomings in RNNs and LSTMs due to vanishing gradients and, consequently, loss of *very long* sequence contexts lead to the transformer model, which introduced self-attention (SA) mechanisms. The SA mechanism enabled focus on different parts of a long input sequence in parallel, which enhanced understanding of contextual nuances in language patterns over extremely long sequences. The SA mechanism is also complemented with positional encodings in transformers to enable the model to maintain an awareness of word/token order, which is required to understand accurate

grammar and syntax. The self-attention mechanism, central to transformers, uses a query, key, and value system to contextualize dependencies in the input sequence. Informally, the SA concept is inspired by classical information retrieval systems where the query is the input sequence context, the key refers to a "database" contained within the parametric memory, and the value is the actual value present at that reference. The operation is mathematically expressed in Equation 1.

$$\text{Attention}(Q, K, V) = \text{softmax}\left(\frac{QK^T}{\sqrt{d_k}}\right)V \qquad (1)$$

In this equation, $Q$, $K$, and $V$ denote the query, key, and value matrices. The scaling factor $\sqrt{d_k}$, where $d_k$ is the dimension of the keys, is employed to standardize the vectors to unit variance for ensuring stable softmax gradients during training. Since the introduction of LLMs with self-attention, there have been several architectural variants depending on the downstream tasks.

**Causal Language Modeling (CLMs)**: CLMs, such as GPT-4, are designed for tasks where text generation is sequential and dependent on the preceding context. They predict each subsequent word based on the preceding words, modeling the probability of a word sequence in a forward direction. This process is mathematically formulated as shown in Equation 2.

$$P(T) = \prod_{i=1}^{n} P(t_i | t_{<i}) \qquad (2)$$

In this formulation, $P(t_i | t_{<i})$ represents the probability of the $i$-th token given all preceding tokens, $t_{<i}$. This characteristic makes CLMs particularly suitable for applications

like content generation, where the flow and coherence of the text in the forward direction are crucial.

**Masked Language Modeling (MLMs)**: Unlike CLMs, MLMs like BERT are trained to understand the bidirectional context by predicting words randomly masked in a sentence. This approach allows the model to learn both forward and backward dependencies in language structure. The MLM prediction process can be represented as Equation 3.

$$P(T_{\text{masked}}|T_{\text{context}}) = \prod_{i \in M} P(t_i|T_{\text{context}}) \quad (3)$$

Here, $T_{\text{masked}}$ is the set of masked tokens in the sentence, $T_{\text{context}}$ represents the unmasked part of the sentence, and $M$ is the set of masked positions. MLMs have proven effective in NLP tasks such as sentiment analysis or question answering.

**Sequence-to-Sequence (Seq2Seq) Modeling**: Seq2Seq models, like T5, are designed to transform an input sequence into a related output sequence. They are often employed in tasks that require a mapping between different types of sequences, such as language translation or summarization. The Seq2Seq process is formulated as Equation 4.

$$P(T_{\text{output}}|T_{\text{input}}) = \prod_{i=1}^{m} P(t_{\text{output}_i}|T_{\text{input}}, t_{\text{output}_{<i}}) \quad (4)$$

In Equation 4, $T_{\text{input}}$ is the input sequence, $T_{\text{output}}$ is the output sequence, and $P(t_{\text{output}_i}|T_{\text{input}}, t_{\text{output}_{<i}})$ calculates the probability of generating each token in the output sequence, considering both the input sequence and the preceding tokens in the output sequence.

In addition to their architectural variants, the utility of LLMs is further enhanced by specific model utilization strategies, enabling their effective adaptation to various domains at scale. One key strategy is fine-tuning, which applies to pre-trained LLMs. Pre-trained LLMs are models already trained on large datasets to understand and generate language, acquiring a broad linguistic knowledge base. Fine-tuning involves further training pre-trained LLMs on a smaller, task-specific dataset, thereby adjusting the neural network weights for particular applications. This process is mathematically represented in Equation 5.

$$\theta_{\text{fine-tuned}} = \theta_{\text{pre-trained}} - \eta \cdot \nabla_\theta L(\theta, D_{\text{task}}) \quad (5)$$

Here, $\theta_{\text{fine-tuned}}$ are the model parameters after fine-tuning, $\theta_{\text{pre-trained}}$ are the parameters obtained from pre-training, $\eta$ is the learning rate, and $\nabla_\theta L(\theta, D_{\text{task}})$ denotes the gradient of the loss function $L$ with respect to the parameters $\theta$ on the task-specific dataset $D_{\text{task}}$.

$$P(T|C) = \prod_{i=1}^{n} P(t_i|t_{<i}, C) \quad (6)$$

Complementing the fine-tuning approach is in-context learning, an alternative strategy that is particularly characteristic of models like the GPT series. This method diverges from fine-tuning by enabling the model to adapt its

responses based on immediate context or prompts without necessitating further training. The efficacy of in-context learning is a direct consequence of the comprehensive pre-training phase, where models are exposed to diverse textual datasets, thereby acquiring a nuanced understanding of language and context. Given a context $C$, the model generates text $T$ that is contextually relevant, as shown in Equation 6. Here, $P(T|C)$ is the probability of generating text $T$ given the context $C$, and $P(t_i|t_{<i}, C)$ is the probability of generating the $i$-th token $t_i$ given the preceding tokens $t_{<i}$ and the context $C$.

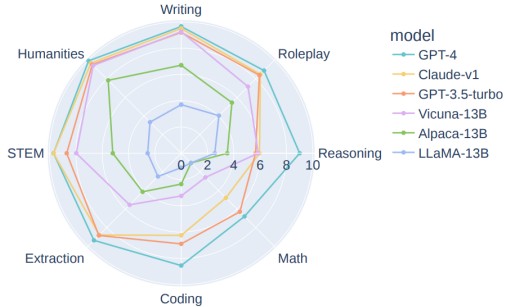

Figure 1: Radar chart showcasing the relative performance of six language models (GPT-4, Claude-v1, GPT-3.5-turbo, Vicuna-13B, Alpaca-13B, LLama-13B) across key domains: Writing, Roleplay, Reasoning, Math, Coding, Extraction, STEM, and Humanities from Zheng et al. (2023a).

These diverse model types and training methodologies under the umbrella of LLMs showcase the flexibility and adaptability of language models in handling a wide range of complex tasks. Figure 1 illustrates the comparative capabilities of different LLMs across various competency domains, such as Writing (evaluating text generation quality), Roleplay (assessing conversational interaction), Reasoning (logical problem-solving), Math (numerical problem-solving), Coding (programming language understanding and generation), Extraction (information retrieval from text), STEM (proficiency in scientific and technical contexts), and Humanities (engagement with arts, history, and social sciences content). Across these domains, GPT-4 exhibits the strongest performance in the benchmark dataset evaluated by Zheng et al. (2023a), indicative of its superior training and extensive knowledge base. Expanding LLMs into applications such as code generation signifies their adaptability and potential for cross-disciplinary innovation. However, fine-tuning and in-context learning methodologies also bring challenges, such as potential data overfitting and reliance on the quality of input context. LLMs' continuous development and refinement promise to open new frontiers in various domains, including automated planning and scheduling, by bridging AI with human-like language understanding.

## Automated Planning and Scheduling

APS is a branch of AI that focuses on the creation of strategies or action sequences, typically for execution by intelligent agents, autonomous robots, and unmanned ve-

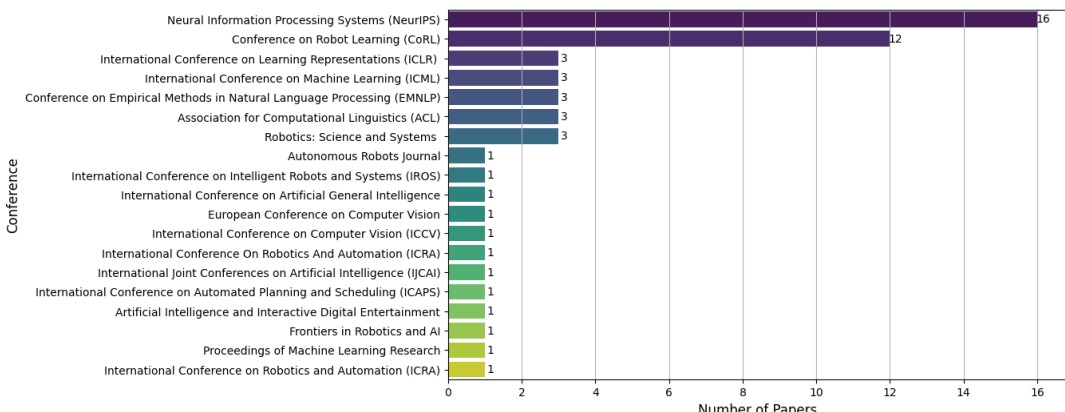

Figure 2: Of the 126 papers surveyed in this study, 55 were accepted by peer-reviewed conferences. This chart illustrates the distribution of these papers across various conferences in the fields of LLMs and APS, highlighting the primary forums for scholarly contributions in these areas.

hicles. A basic category in APS is a Classical Planning Problem (CPP) (Russell and Norvig 2003) which is a tuple $\mathcal{M} = \langle \mathcal{D}, \mathcal{I}, \mathcal{G} \rangle$ with domain $\mathcal{D} = \langle F, A \rangle$ - where $F$ is a set of fluents that define a state $s \subseteq F$, and $A$ is a set of actions - and initial and goal states $\mathcal{I}, \mathcal{G} \subseteq F$. Action $a \in A$ is a tuple $(c_a, pre(a), eff^{\pm}(a))$ where $c_a$ is the cost, and $pre(a), eff^{\pm}(a) \subseteq F$ are the preconditions and add/delete effects, i.e., $\delta_{\mathcal{M}}(s, a) \models \perp s \ if \ s \not\models pre(a)$; $else \ \delta_{\mathcal{M}}(s, a) \models s \cup eff^+(a) \setminus eff^-(a)$ where $\delta_{\mathcal{M}}(\cdot)$ is the transition function. The cumulative transition function is $\delta_{\mathcal{M}}(s, (a_1, a_2, \ldots, a_n)) = \delta_{\mathcal{M}}(\delta_{\mathcal{M}}(s, a_1), (a_2, \ldots, a_n))$. A plan for a CPP is a sequence of actions $\langle a_1, a_2, \ldots, a_n \rangle$ that transforms the initial state $\mathcal{I}$ into the goal state $\mathcal{G}$ using the transition function $\delta_{\mathcal{M}}$. Traditionally, a CPP is encoded using a symbolic representation, where states, actions, and transitions are explicitly enumerated. This symbolic approach, often implemented using Planning Domain Definition Language or PDDL (McDermott et al. 1998), ensures precise and unambiguous descriptions of planning problems. This formalism allows for applying search algorithms and heuristic methods to find a sequence of actions that lead to the goal state, which is the essence of the plan.

The advent of LLMs has sparked a significant evolution in representation methods for CPPs, moving towards leveraging the expressive power of natural language (Valmeekam et al. 2023a) and the perceptual capabilities of vision (Asai 2018). These novel approaches, inherently more suited for LLM processing, use text and vision-based representations, allowing researchers to utilize the pre-existing knowledge within LLMs. This shift enables a more humanistic comprehension and reasoning about planning tasks, enhancing the flexibility and applicability of planning algorithms in complex, dynamic environments. LLMs, while distinct in being trained on vast datasets outside the traditional scope of planning, loosely connect to previous data-driven methodologies, such as case-based reasoning (Xu 1995) applied to planning and Hierarchical Task Network (HTN) (Georgievski and Aiello 2015) which make use of task knowledge. It is an open area how LLMs may be used synergestically with prior methods.

## LLMs in APS – Literature selection

A comprehensive survey of existing literature was conducted to explore the application of LLMs for automated planning. This endeavor led to identifying 126 pertinent research papers showcasing various methodologies, applications, and theoretical insights into utilizing LLMs within this domain.

The selection of these papers was guided by stringent criteria, focusing primarily on their relevance to the core theme of LLMs in automated planning. The search, conducted across multiple academic databases and journals, was steered by keywords such as "Large Language Models", "Automated Planning", "LLMs in Planning", and "LLMs + Robotics". Figure 2 presents the distribution of these selected papers across various peer-reviewed conferences, underlining the breadth and diversity of forums addressing the intersection of LLMs and APS. Even if a paper originated from a workshop within a conference, only the conference name is listed. Out of 126 papers, 71 are under review or available on arXiv. The inclusion criteria prioritized the relevance and contribution of papers to automated planning with LLMs over the publication date. Nonetheless, all surveyed papers emerged from either 2022 or 2023, a trend depicted in Figure 3, underscoring the recent surge in LLM research. A word cloud was generated to visually capture the prevalent research themes reflected in these papers' titles, illustrated in Figure 4. This cloud highlights the frequent use of terms such as "Language Model" and "Planning", which dominate the current discourse. In contrast, the emergence of "Neuro-Symbolic" reflects a nascent yet growing interest in integrating neural and symbolic approaches within the field. This systematic approach ensured a comprehensive inclusion of seminal works and recent advancements.

Upon the accumulation of these papers, a meticulous manual categorization was undertaken. The papers were divided into four piles, each containing approximately 30 papers. Each pile was manually categorized by one author,

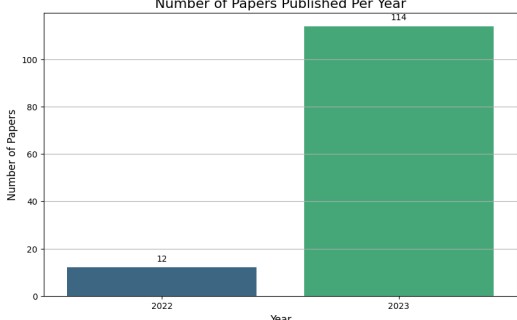

Figure 3: Annual distribution of the 126 surveyed papers, indicating a significant increase in publications from 12 in 2022 to 114 in 2023, highlighting the rapid growth of LLM research within a single year.

with the final categorization being reviewed by all authors. During this process, each paper could belong to multiple categories out of the eight established. The maximum number of categories assigned to a single paper was three, although the median was typically one category per paper. This process was pivotal in distilling the vast information into coherent, thematic groups. The categorization was conducted based on the specific application of LLMs in planning. This formed eight distinct categories, each representing a unique facet of LLM application in automated planning. These categories facilitate a structured analysis and highlight LLMs' diverse applications and theoretical underpinnings in this field.

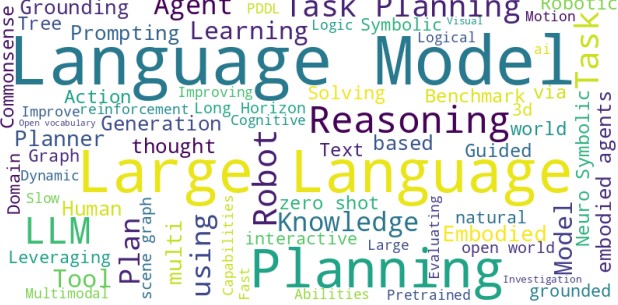

Figure 4: Word cloud of terms from the titles of papers surveyed in this study, displaying the prevalence of "Language Model" and "Planning" as central themes. The presence of "Neuro-Symbolic" indicates an emergent trend toward the fusion of neural and symbolic methodologies in the domain.

## LLMs in APS – Literature Discussion

This section dwelves into the diverse applications of LLMs in planning tasks. We have identified eight distinct categories based on the utility and application of LLMs in planning, which are concisely summarized in Table 1. Figure 5 provides a detailed taxonomy, illustrating the categorization of the identified research papers.

## Language Translation

Language translation in the context of LLMs and planning involves transforming natural language instructions into structured planning languages (Wong et al. 2023; Kelly et al. 2023; Yang 2023; Pan et al. 2023; Xie et al. 2023; Yang, Ishay, and Lee 2023; Lin et al. 2023c; Sakib and Sun 2023; Yang et al. 2023b; Parakh et al. 2023; Yang et al. 2023a; Dai et al. 2023; Ding et al. 2023b; Zelikman et al. 2023; Xu et al. 2023b; Chen et al. 2023a; You et al. 2023) such as PDDL, and vice versa, utilizing in-context learning techniques (Guan et al. 2023). This capability effectively bridges the gap between human linguistic expression and machine-understandable formats, enhancing intuitive and efficient planning processes. The LLM+P framework (Liu et al. 2023) exemplifies this by converting natural language descriptions of planning problems into PDDL using GPT-4, leveraging classical planners for solution finding, and then translating these solutions back into natural language, with a specific focus on robot planning scenarios. Additionally, Graph2NL (Chalvatzaki et al. 2023) generates natural language text from scene graphs for long-horizon robot reasoning tasks, while (Shirai et al. 2023) introduces a vision-to-language interpreter for robot task planning. Further, (Brohan et al. 2023) examines the grounding of LLM-generated natural language utterances in actionable robot tasks, and (Yang, Gaglione, and Topcu 2022) utilizes LLMs for creating finite-state automatons for sequential decision-making problems. Despite these advancements, a critical research gap emerges in the autonomous translation capabilities of LLMs, particularly in converting natural language to PDDL without external expert intervention.

> While LLMs effectively translate PDDL to natural language, **a notable gap is evident in their limited understanding of real-world objects and the problem of grounding affordances**, mainly when translating natural language to structured languages like PDDL. Addressing this gap calls for integrating neuro-symbolic approaches in LLMs, where the fusion of perceptual experience for concrete concept understanding from knowledge graphs complements LLMs' proficiency in distributional statistics (Lenat and Marcus 2023).

## Plan Generation

This category focuses on directly generating plans using LLMs. The research, primarily utilizing causal language models through in-context learning (Sermanet et al. 2023; Li et al. 2023b; Silver et al. 2023; Parakh et al. 2023; Zelikman et al. 2023; Besta et al. 2023; Huang et al. 2023a; Dalal et al. 2023; Wang et al. 2023b; Valmeekam et al. 2022; Valmeekam, Marquez, and Kambhampati 2023; Gramopadhye and Szafir 2022; Singh et al. 2023)[1], demonstrates modest performance, indicating notable challenges in employing LLMs for effective plan generation. Novel in-context learning strategies, such as the Chain-of-Symbol and Tree of Thoughts, have been introduced to enhance LLMs' reasoning capabilities (Hu et al. 2023b; Yao et al. 2023). Ef-

---

[1]Due to space constraints, only a select number of papers are cited in this section.

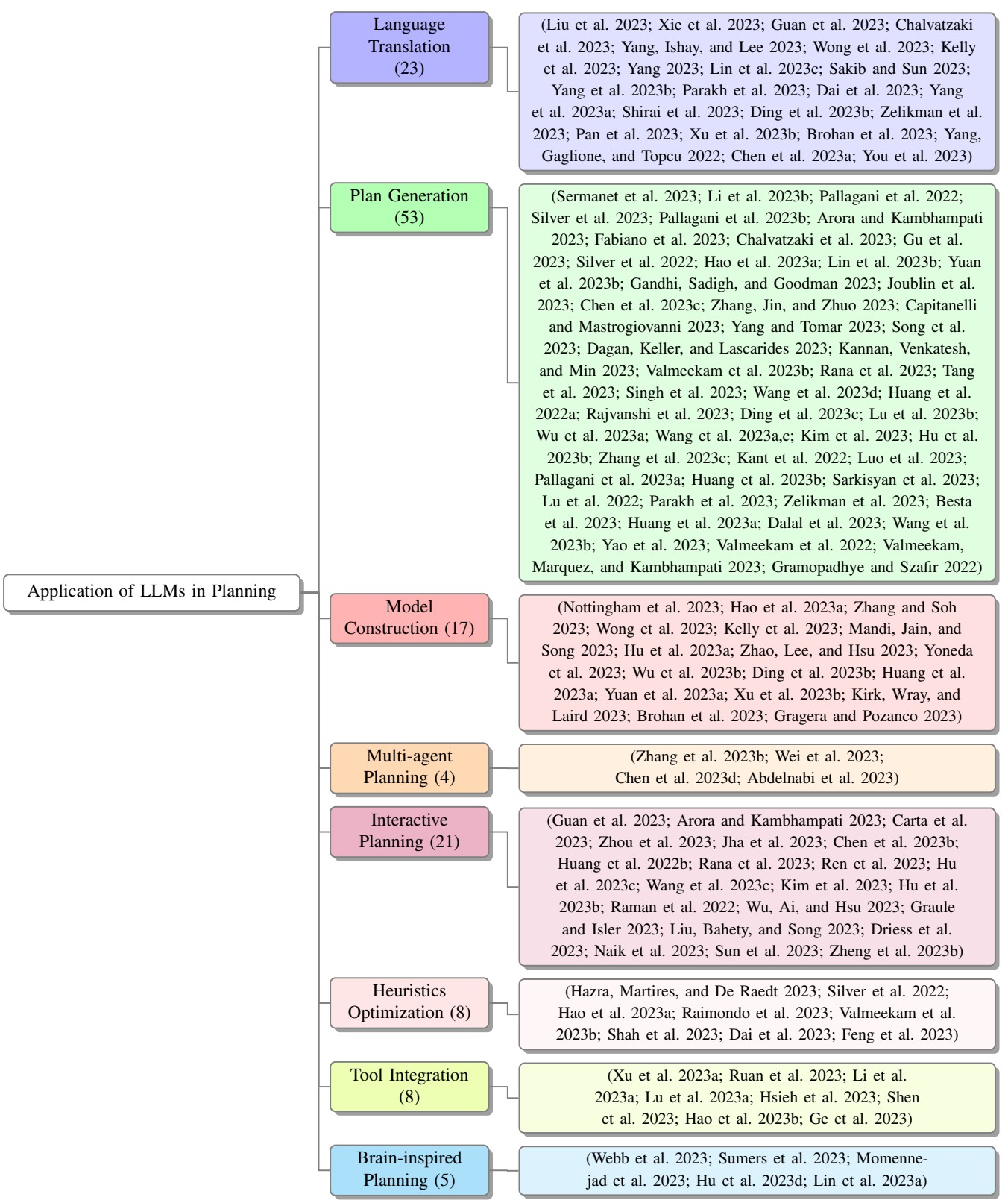

Figure 5: Taxonomy of recent research in the intersection of LLMs and Planning into categories (#). Each has scholarly papers based on their unique application or customization of LLMs in addressing various aspects of planning problems.

forts to generate multimodal, text, and image-based goal-conditioned plans are exemplified by (Lu et al. 2023b). Additionally, a subset of studies in this survey investigates the fine-tuning of seq2seq, code-based language models (Pallagani et al. 2022, 2023b), which are noted for their advanced syntactic encoding. These models show promise in improving plan generation within the confines of their training datasets (Logeswaran et al. 2023), yet exhibit limitations in generalizing to out-of-distribution domains (Pallagani et al. 2023a), highlighting a gap in their adaptability across diverse planning contexts.

> Causal LLMs are predominantly used for plan generation, but their performance is often **limited due to their design, which is focused on generating text based on preceding input.** On the other hand, seq2seq LLMs can generate valid plans but **struggle with generalization across diverse domains.** This limitation highlights an opportunity for a synergistic approach: integrating even imperfect LLM outputs with symbolic planners can expedite heuristic searches, thereby enhancing efficiency and reducing search times (Fabiano et al. 2023).

### Model Construction

This category employs LLMs to build or refine world and domain models essential for accurate planning. Nottingham et al. (2023); Yuan et al. (2023a) leverage in-context learning with LLMs to develop an abstract world model in the Minecraft domain, highlighting the challenge of semantic grounding in LLMs. Similarly, Gragera and Pozanco (2023) explore the capability of LLMs in completing ill-defined PDDL domains. Efforts such as (Huang et al. 2023a; Brohan et al. 2023) delve into LLMs' grounding capabilities, with SayCan (Brohan et al. 2023) notably achieving 74% executable plans. Hao et al. (2023a); Yoneda et al. (2023) innovatively positions LLMs as both world models and reasoning agents, enabling the simulation of world states and prediction of action outcomes. Research by (Zhang and Soh 2023; Wong et al. 2023; Mandi, Jain, and Song 2023; Hu et al. 2023a; Zhao, Lee, and Hsu 2023; Ding et al. 2023b; Huang et al. 2023a; Wu et al. 2023b; Xu et al. 2023b; Brohan et al. 2023) shows that LLMs can effectively model high-level human states and behaviors using their commonsense knowledge. Yet, they face difficulties accurately processing low-level geometrical or shape features due to spatial and numerical reasoning constraints. Additionally, Kelly et al. (2023) investigates the potential of LLMs in conjunction with planners to craft narratives and logical story models, integrating human-in-the-loop for iterative edits.

> LLMs often **struggle with detailed spatial reasoning and processing low-level environmental features, limiting their effectiveness in model construction.** Integrating world models presents a viable solution, offering advanced abstractions for reasoning that encompass human-like cognitive elements and interactions, thereby enhancing LLMs' capabilities in model construction (Hu and Shu 2023).

### Multi-agent Planning

In multi-agent planning, LLMs play a vital role in scenarios involving interaction among multiple agents, typically modeled using distinct LLMs. These models enhance coordination and cooperation, leading to more complex and effective multi-agent strategies. (Zhang et al. 2023b) introduces an innovative framework that employs LLMs to develop cooperative embodied agents. AutoGraph (Wei et al. 2023) leverages LLMs to generate autonomous agents adept at devising solutions for varied graph-structured data problems. Addressing scalability in multi-robot task planning, (Chen et al. 2023d) proposes frameworks for the collaborative function of different LLM-based agents. Furthermore, (Abdelnabi et al. 2023) explores the effectiveness of LLM agents in complex, text-based, multi-agent negotiation games, revealing their capacity to negotiate and consistently reach successful agreements.

> A key gap in multi-agent planning with LLMs lies in **standardizing inter-agent communication and maintaining distinct belief states, including human aspects.** Overcoming this requires advanced LLM algorithms for dynamic alignment of communication and belief states, drawing on epistemic reasoning and knowledge representation (de Zarzà et al. 2023).

### Interactive Planning

In this category, LLMs are utilized in dynamic scenarios where real-time adaptability to user feedback or iterative planning is essential. The refinement of LLM outputs is typically achieved through four primary feedback variants: **(a)** External verifiers, such as VAL(Howey, Long, and Fox 2004) for PDDL or scene descriptors and success detectors in robotics (Guan et al. 2023; Arora and Kambhampati 2023; Jha et al. 2023; Huang et al. 2022b; Liu, Bahety, and Song 2023; Rana et al. 2023; Ren et al. 2023; Kim et al. 2023; Graule and Isler 2023; Driess et al. 2023; Zheng et al. 2023b); **(b)** Online reinforcement learning, which progressively updates the LLM about environmental changes (Carta et al. 2023); **(c)** Self-refinement by LLMs, where they provide feedback on their own outputs (Zhou et al. 2023; Hu et al. 2023c,b; Ding et al. 2023a; Sun et al. 2023; Naik et al. 2023); **(d)** Input from human experts (Raman et al. 2022; Wu, Ai, and Hsu 2023). Furthermore, (Chen et al. 2023b) introduces the "Action Before Action" method, enabling LLMs to proactively seek relevant information from external sources in natural language, thereby improving embodied decision-making in LLMs by 40%.

> A key gap in interactive planning with LLMs lies in harmonizing the "fast" neural processing of LLMs with "slow" symbolic reasoning, as manifested in feedback mechanisms. This integration is key to **maintaining the neural speed of LLMs while effectively embedding the depth and precision of feedback**, which is vital for accuracy in dynamic planning scenarios (Zhang et al. 2023a).

### Heuristics Optimization

In the realm of Heuristics Optimization, LLMs are leveraged to enhance planning processes, either by refining existing plans or aiding symbolic planners with heuristic guidance. Studies like (Hazra, Martires, and De Raedt 2023; Hao et al. 2023a; Dai et al. 2023; Feng et al. 2023) have effectively

coupled LLMs with heuristic searches to identify optimal action sequences. Research by (Silver et al. 2022; Shah et al. 2023; Valmeekam et al. 2023b) reveals that LLMs' outputs, even if partially correct, can provide valuable direction for symbolic planners such as LPG (Gerevini and Serina 2002), especially in problems beyond the LLMs' solvable scope. Furthermore, (Raimondo et al. 2023) makes an intriguing observation that including workflows and action plans in LLM input prompts can notably enhance task-oriented dialogue generalization.

> This category marks significant progress towards realizing neuro-symbolic approaches in APS. **Current methods emphasize plan validity, often at the expense of time efficiency.** Future research should look at how to continually evolve LLMs for better plan generation, with its experience from complimenting symbolic planners (Du et al. 2023).

## Tool Integration

In tool integration, LLMs serve as coordinators within a diverse array of planning tools, enhancing functionality in complex scenarios. Studies like (Xu et al. 2023a; Lu et al. 2023a; Shen et al. 2023; Hao et al. 2023b; Ge et al. 2023) demonstrate that incorporating tools such as web search engines, Python functions, and API endpoints enhances LLM reasoning abilities. However, (Ruan et al. 2023) notes a tendency for LLMs to over-rely on specific tools, potentially prolonging the planning process. (Li et al. 2023a) introduces a benchmark for tool-augmented LLMs. While typical approaches involve teaching LLMs tool usage via multiple prompts, (Hsieh et al. 2023) suggests that leveraging tool documentation offers improved planning capabilities, circumventing the need for extensive demonstrations.

> LLMs often **hallucinate non-existent tools, overuse a single tool, and face scaling challenges with multiple tools.** Overcoming these issues is key to enabling LLMs to effectively select and utilize various tools in complex planning scenarios (Elaraby et al. 2023).

## Brain-Inspired Planning

This area explores neurologically and cognitively inspired architectures in LLMs (Webb et al. 2023; Sumers et al. 2023; Momennejad et al. 2023; Hu et al. 2023d; Lin et al. 2023a), aiming to replicate human-like planning in enhancing problem-solving. However, while these methods rely on in-context learning, they frequently encounter challenges such as hallucination and grounding, as previously discussed, and tend to be more computationally intensive than in-context learning alone.

> While LLMs attempt to mimic symbolic solvers through in-context learning for brain-inspired modules, this approach **lacks adaptability and is a superficial understanding of complex cognitive processes.** To overcome these issues, developing systems where neural and symbolic components are intrinsically intertwined is critical as it would accurately mirror human cognitive capabilities in planning (Fabiano et al. 2023).

## Discussion and Conclusion

In this position paper, we comprehensively investigate the role of LLMs within the domain of APS, analyzing 126 scholarly articles across eight distinct categories. This extensive survey not only provides a detailed landscape of current LLM applications and their limitations but also highlights the volume of research in each category: Language Translation with 23 papers demonstrates LLMs' proficiency, whereas Plan Generation, the most researched category with 53 papers, reveals their shortcomings in optimality, completeness, and correctness compared to traditional combinatorial planners. Our exploration extends to Model Construction (17 papers), which examines LLMs in developing planning models, and the relatively unexplored area of Multi-agent Planning (4 papers). Interactive Planning is well represented with 21 papers, illustrating LLMs' adaptability in feedback-centric scenarios. Despite being less researched, Heuristics Optimization and Tool Integration, each with 8 papers, provide valuable insights into efficiency enhancement and integration of LLMs with symbolic solvers. Lastly, Brain-inspired Planning, although least represented with 5 papers, opens innovative avenues for human-like planning processes in LLMs. By identifying the research distribution and gaps in these categories, our paper proposes how neuro-symbolic approaches can address these voids, thereby underscoring the varying degrees of LLM applications in APS and guiding future research towards enhancing their capabilities in complex planning tasks.

It is important to acknowledge that while LLMs have shown promise, they are not a panacea for the inherent complexities of automated planning. The expectation that LLMs, operating within polynomial run-time bounds, could supplant the nuanced and often non-polynomial complexities of symbolic planners is not yet realizable. Indeed, the strengths of LLMs do not currently include generating sequences of actions akin to those devised by symbolic planners, which are essential for creating a coherent and practical plan for complex problems. However, this does not diminish the potential utility of LLMs within this space. When considering average-case scenarios, which are typically less complex than worst-case scenarios, LLMs could offer substantial efficiencies. They can be seen as akin to meta-heuristic approaches, capable of accelerating plan generation in a variety of settings. As such, their application, governed by cognitive-inspired frameworks like SOFAI(Fabiano et al. 2023), could delineate when and where their use is most advantageous.

Future research should prioritize three areas: developing new LLM training paradigms that ensure coherence and goal alignment in outputs; delving into Henry Kautz's neuro-symbolic taxonomies (Kautz 2022) to better integrate neural and symbolic methods; and establishing clear performance metrics for LLM-assisted planners. In conclusion, integrating LLMs into automated planning, while challenging, opens avenues for innovation. Embracing a symbiotic approach that combines the creative strengths of LLMs with the precision of symbolic planners can lead to more effective, sophisticated AI applications in planning.

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
