# OpenReview forum: "On the Prospects of Incorporating Large Language Models (LLMs) in Automated Planning and Scheduling (APS)"
_icaps-conference.org/ICAPS/2024/Conference — ICAPS 2024_

### Official Review · Reviewer_cKZE · 2024-01-21

**Significance And Importance:** 3
**Soundness:** 4
**Novelty:** 3
**Clarity:** 4
**Overall Evaluation:** 3
**Confidence:** 5

**Weaknesses:**

2: No major or minor weaknesses.

**Contributions Of The Paper:**

The paper is a mixture of a position paper and a literature survey. It nicely reviews very recent literature and contributions in the field of LLMs for APS, intertwining the review with position statements. These tend to be open-ended and do not hint at solutions but do point out relevant areas for future development and investigation.

**Ethical Considerations:**

(1) Not Applicable: The paper does not have any ethical considerations to address

**Nomination For Best Paper:**

No

**Questions For Authors:**

There has been some work on LLMs for service composition (service composition is basically planning):
1. Have you explored the existing literature on service composition using Large Language Models (LLMs) in your research?
2. How do you think incorporating the concept of service composition, which is closely related to planning, could enhance the scope of your literature review?

**Reproducibility:**

0: N/A - nothing to reproduce.

**Strengths Of The Paper:**

Timely topic
Good literature coverage
Nice discussion of the literature and perspectives opened by recent work

**Weaknesses Of The Paper:**

Open directions are indicated, but no hint is given on how to move forward. There are no major weaknesses.

---

> ### Author Rebuttal · Authors · 2024-01-27
>
> ## **Q. Have you explored the existing literature on service composition, which is closely related to planning, and how could it enhance the scope of the current literature review? _(merged Q1 and Q2)_**
>
> The focus of the current paper is on action-based planning. We recognize that planning has been used extensively in different settings of service composition (like web services, workflows, processes, and APIs). Extending our study to cover LLMs for service composition is an opportunity for future work. Indeed, LLMs can play an important role in advancing service composition. However, services are different from actions in terms of representation, semantics, failure,  and evaluation criteria, to name a few.

---

### Official Review · Reviewer_cADm · 2024-01-22

**Significance And Importance:** 2
**Soundness:** 3
**Novelty:** 1
**Clarity:** 2
**Overall Evaluation:** 1
**Confidence:** 3

**Weaknesses:**

2: No major or minor weaknesses.

**Contributions Of The Paper:**

The paper is a survey of the literature on the usage of LLM for planning. The authors classify a large set of papers into 8 categories: language translation, plan generation, model construction, multi-agent planning, interactive planning, heuristics optimization, tool integration, and brain-inspired planning.

**Ethical Considerations:**

(1) Not Applicable: The paper does not have any ethical considerations to address

**Nomination For Best Paper:**

No

**Questions For Authors:**

(1) How LLMs are used for interactive planning?
(2) Which is the aim of using LLM for tool integration?

**Reproducibility:**

0: N/A - nothing to reproduce.

**Strengths Of The Paper:**

I appreciated that the authors try to draw general conclusions for each of the identified category.

**Weaknesses Of The Paper:**

Some sections are described too quickly. Perhaps, a journal would be a better recipient for this survey, rather than conference proceedings with a limited number of pages. For example, it is not clear to me how LLMs are used for interactive planning. Similarly, it is not clear the aim of using LLM for tool integration.

Minor issues:
Line 218. What does symbol \bot mean here?
Line 438. even if ONLY partially correct
Line 439. As far as I know, LLMs have been used to construct an initial plan that is subsequently adapted by LPG. This usage is not really heuristic optimization. A better name for this category could be "search guidance".
Line 464. It is not clear what human-like planning is.
Line 493. "our paper proposes how..." Really? It seems to me that the paper describes a number of open issues, without proposing solutions.

---

> ### Author Rebuttal · Authors · 2024-01-27
>
> ## **Q. How are LLMs used for interactive planning?**
>
> LLMs were used for interactive planning by engaging in a multi-turn dialogue process like in [1] (already in the paper). This approach uses LLMs for multiple turns (often denoted as "n-turns") until a desired answer or solution is obtained. This interactive process involves conversations between the user or external verifiers/validators and the LLM.
>
> [1] Wang, Zihao, et al. "Describe, explain, plan and select: Interactive planning with large language models enables open-world multi-task agents." arXiv preprint arXiv:2302.01560 (2023).
>
> ## **Q. What is the aim of using LLMs for tool integration?**
>
> An LLM can improve the tools’ interoperability and functionality through its advanced natural language processing and generation capabilities. Moreover, an LLM can become an agent, thus impacting the real world through the tools it orchestrates. For example, [2] (already in the paper) uses an LLM with 12 different tools, such as a Python generator and shell generator, and uses an LLM to optimize the tools’ usage in task planning.
>
> [2] Ruan, Jingqing, et al. "Tptu: Task planning and tool usage of large language model-based ai agents." arXiv preprint arXiv:2308.03427 (2023).
>
> We will take care of the minor issues mentioned by the reviewer in the final version of the paper.

---

### Official Review · Reviewer_1Ctf · 2024-01-23

**Significance And Importance:** 3
**Soundness:** 4
**Novelty:** 3
**Clarity:** 3
**Overall Evaluation:** 3
**Confidence:** 3

**Weaknesses:**

2: No major or minor weaknesses.

**Contributions Of The Paper:**

In this position paper authors extensively explore the litterature on different uses, applications and theoretical prospects of Large Language Models (LLM) for Automated Planning and Scheduling (APS). The study comprises a corpus of 126 papers from 2022 and 2023.
The authors extracted from this corpus 8 main topics regarding LLM for APS: Language translation, Plan generation, Model construction, Multi-agent planning, Interactive planning, Heurictic optimization, Tool interaction and Brain inspired learning. For each category the authors provide current achievements and key limitation to overcome for a better combination of LLM and APS.
More specifically the authors highlight that while LLM may look like a promising avenue to tackle the inherently non-polinomial tasks that arise in APS, they lack the precision to correctly and rigorously handle such tasks. However, LLM do provide valuable help and insights that can be used by symbolic systems.
From this constatation the authors advocate for research in neuro-symbolic approaches where neural-based methods work in tandem with symbolic planners.

**Ethical Considerations:**

(1) Not Applicable: The paper does not have any ethical considerations to address

**Nomination For Best Paper:**

No

**Questions For Authors:**

Q1. In response to the weakness mentionned above, what would be, in our current knowledge about how LLM and APS interact, the most relevant form of neuro-symbolic approach for planning ?

**Reproducibility:**

0: N/A - nothing to reproduce.

**Strengths Of The Paper:**

This paper is clear, and conveys its message eloquently. Obviously the sheer volume of recent papers analysed to support the position show how this topic is of utmost importance in the community but also that this position is well supported by the current state of the research. Finally the way the litterature was analysed was very insightful. The main topics were derived from actual research and not decided beforehand and the analysis on the volume in each category shows why such a position paper is of importance to redirect some effort towards more promising avenues.

**Weaknesses Of The Paper:**

Although the authors advocate for neuro-symbolic approaches, this landscape is vast and varied. It does seem that the authors point towards a real collaboration of LLM and symbolic planners (so stepping away from the "Symbolic Neuro symbolic" taxon as described in Kautz 2022) but this collaboration can still take many forms and additional insights would have been appreciated.

---

> ### Author Rebuttal · Authors · 2024-01-27
>
> ## **Q. What is the most relevant form of neuro-symbolic approach for planning?**
>
> There are indeed many ways to integrate LLMs and APS, and not all of them fall under Henry Kautz’s taxonomy [1]. The one we think is most promising is SOFAI (for Slow and Fast AI) [2] (already in the paper) - a multi-agent cognitive architecture inspired by Kahneman's "Thinking Fast and Slow" theory of human reasoning and decision-making. The architecture is designed to be modular as to enable the incorporation of diverse "fast" (S1) and "slow" (S2) solvers, decision environments, score models (as reward, risk and value alignment), agent deployment economy (solution cost complexity), and emergent behavior attributes. Applied to planning, LLM-based planners can be considered S1solvers and APS can be considered S2 solvers, orchestrated together with a metacognition module.
>
> [1] Kautz, Henry. "The Third AI Summer: AAAI Robert S. Engelmore Memorial Lecture." AI Magazine 43.1 (2022): 105-125.
>
> [2] SOFAI - https://sites.google.com/view/sofai/publications. In particular,
> a) Francesco Fabiano, Vishal Pallagani, Marianna Bergamaschi Ganapini, Lior Horesh, Andrea Loreggia, Keerthiram Murugesan, Francesca Rossi, Biplav Srivastava. Plan-SOFAI: A Neuro-Symbolic Planning Architecture. NuCLeaR Workshop at AAAI 2024.
> b) Grady Booch, Francesco Fabiano, Lior Horesh, Kiran Kate, Jonathan Lenchner, Nick Linck, Andrea Loreggia, Keerthiram Murugesan, Nicholas Mattei, Francesca Rossi, Biplav Srivastava. Thinking Fast and Slow in AI. AAAI 2021: 15042-15046

---

### Meta-Review · Area_Chair_1fer · 2024-02-01

**Recommendation:** Accept (Oral)
**Confidence:** 5

**Metareview:**

Clear case. Congrats!

**Ethical Considerations:**

(4) Good: The paper adequately addresses most, but not all, of the applicable ethical considerations